# An Optimization Method for Equalizing the Spatial Accessibility of Medical Services in Guangzhou

**Mingkai Yu** [1][iD]**, Yingchun Fu** [1,*] **and Wenkai Liu** [2]

1   School of Geography, South China Normal University, Guangzhou 510631, China;
   2021022885@m.scnu.edu.cn
2   Beidou Research Institute, Faculty of Engineering, South China Normal University, Guangzhou 510631, China;
   wenkai.liu@m.scnu.edu.cn
*   Correspondence: fuyc@m.scnu.edu.cn

**Abstract:** Spatial equality of medical services refers to equal access to medical services in all regions. Currently, research on medical facility planning focuses mainly on efficiency, and less on methods for achieving medical facility access equality. In this study, we propose a medical service equality optimization method considering facility grade and Gaode actual travel time data. First, we use the maximum coverage location problem (MCLP) model to locate new medical facilities. Then, we incorporate a service capacity weight matrix reflecting medical facility grade into the quadratic programming (QP) model, with the objective of optimizing the bed configuration of each facility to maximize the spatial equality of medical accessibility. By measuring and optimizing medical accessibility in Guangzhou under different travel time thresholds, we analyzed the optimization results of central, peripheral, and edge areas. The results show that (1) the model significantly improves the spatial equality of medical accessibility. After optimization, fewer locations have very low (or low) and very high (or high) accessibility, while more locations have moderate accessibility. When the travel time threshold is 22 min, the number of locations with medium accessibility level increases by about 18.86%. (2) The higher the travel time threshold, the greater is the overall optimization effect. (3) Different regions have different optimization effects and a larger travel time threshold can improve the optimization effect of the peripheral areas more significantly. It is recommended that new medical facilities be built in the peripheral and edge areas, along with improvements to the transport system.

**Keywords:** medical accessibility; spatial equality; maximum covering location problem (MCLP); quadratic programming (QP); medical facility grade

## 1. Introduction

Spatial inequalities in access to medical services across regions are a major and persisting challenge for countries around the world [1–6]. As the largest developing country in the world, China has significant spatial differences in access to medical services [7,8]. In the 14th Five-Year Plan released by the Chinese government in 2021, it was emphasized that health care services should be more accessible and that the gaps between different groups and regions should be gradually narrowed. This policy embodies two important concepts in health care planning, "efficiency" and "equality". "Efficiency" emphasizes maximum coverage or quantity of medical services at minimal travel cost, while "equality" emphasizes equal access to medical services for all groups or regions. However, "efficiency" and "equality" are often competing goals in urban areas and are difficult to achieve simultaneously, especially in areas with significant medical disparities between the city center and suburbs. For a long time, people have been discussing and trying to make a choice between them in the planning of medical service facilities [9–12].

However, most research on health service planning has focused on efficiency. For example, traditional location allocation models, such as the location set coverage problem (LSCP) [13], maximum coverage location problem (MCLP) [14], p-median problem (p-median) [15], and many improved algorithms, such as the double standard model (DSM) [16], the maximum expected coverage location problem (MEXCLP) [17], the maximum available location problem (MALP) [18], the dynamic available coverage location model (DACL) [19], have been developed for this purpose. These models improve the efficiency of accessing medical services by optimizing one or more of the following objectives: minimizing travel costs, minimizing resource investment, and maximizing demand coverage. These models are widely used in medical service planning as they can be easily implemented through existing GIS software. In contrast, the equality of access to medical services has received less attention because it is extremely difficult to achieve among all regions or population groups [20] and there is a lack of integrated toolkits in current GIS software, thus making complex mathematical modeling and programming necessary. These reasons have contributed to the slow research progress aimed at reducing regional disparities by addressing equality in health care.

Spatial inequalities in medical services are mainly reflected in spatial differences in the number of available health care services, which reflect regional differences in the accessibility of medical services [21,22]. Accessibility, a concept borrowed from the field of transportation planning, has also been widely used in the field of urban public services [23–26]. In the field of medical services, accessibility is often used to measure how easy it is for people to access medical services [27–29]. There are many methods for measuring spatial accessibility, including (1) proximity [30], (2) cumulative chance [31,32], (3) the gravity model [33,34], and (4) the two-step floating catchment area (2SFCA). Among them, 2SFCA and its improved model have become the most common methods for measuring spatial accessibility as they take into full consideration the relationship between supply and demand and the distance decay effect. Currently, the improvements on the 2SFCA model mainly focus on catchment area delineation [35,36], decay function [37,38], travel mode [39,40], measurement of supply capacity [41], and supply and demand ratio [42]. These models improve the 2SFCA model by different degrees according to various application requirements, and greatly expand the application scenarios of the 2SFCA model.

In recent years, medical spatial inequality based on accessibility has gradually attracted people's attention [43]. Spatial equality measurements include maximum deviation (MD), coefficient of variance (CV), standard deviation (SD), Theil index, and Gini coefficient (GC). Yin et al. [43] used the Theil Index to evaluated health resource inequality in 2859 counties in China. Lu et al. [44] used CV to compare the fairness of medical resources in Beijing before and after the referral reform. Xia et al. [45] used the GC to explore how traffic conditions at various times affect the equality of access to medical services. Tao and Han [46] used GC and CV to evaluate the inequality of medical service accessibility in Shenzhen under the hierarchical medical system. Chen et al. [47] used GC to evaluate the impact of different searching thresholds on the equality of medical services. Similar methods have also been applied to evaluate the equality of access to other social public services, such as education [48], community green space [49,50], transportation [51], etc.

Although a large number of studies have focused on using various indicators to evaluate the inequality of existing medical services and the causes of inequality, only a few have formulated specific optimization plans for overcoming the inequality of medical services. Studies have indicated that optimizing the equality of medical services should include determining the optimal location and capacities of service facilities [52,53]. For example, Wang and Tang [20] introduced the quadratic programming (QP) method into the redistribution of medical resources to achieve maximum equality in medical services. Tao et al. [54] used particle swarm optimization (PSO) to achieve maximum equality in the accessibility of elderly care services in Beijing. Tao et al. [55] also used particle swarm optimization (PSO) to optimize the medical accessibility of hospitals at different levels

in Shenzhen. Li et al. [53] optimized EMS accessibility in downtown Shanghai in three scenarios from the perspective of trade-offs.

Although existing research has laid a foundation for improving medical service equality, there are still gaps in the following areas: (1) very few accessibility-based medical equality optimization models consider facility grade differences; (2) there is no reasonable limit in the optimization range of facility capacity, resulting in the possible situation of zero capacity; (3) there is a lack of exploration on how different travel time thresholds affect the optimization effects and their influencing factors. Moreover, the optimized model and result need to be assessed in a typical city with the inequality state of art of medical services.

Guangzhou is a first-tier city in China with abundant medical resources and high-quality medical services. However, different regions vary in their access and use of these resources. Based on the "Guangzhou Health Development Statistical Bulletin" published by the Guangzhou Municipal Health Commission in 2018, older districts such as Yuexiu District, Haizhu District, and Liwan District have more Grade III hospitals and specialty hospitals than suburban districts such as Conghua District and Zengcheng District which lack advanced medical facilities. Moreover, due to a better transportation infrastructure and an extensive rail transit network, it is easier and faster for residents in central districts to reach various types of medical institutions than those living in remote districts who face transportation difficulties and longer travel times. Overall, because of the unequal spatial distribution of medical resources and different degrees of transport accessibility, there are significant spatial disparities in medical accessibility in Guangzhou. How to improve the spatial equality of medical accessibility is a worthwhile research question.

To improve the spatial inequality of medical accessibility caused by the difference in the distribution of medical resources, this study implemented an optimization model of medical accessibility that considers the grade and the capacities of facilities in Guangzhou. We optimized the spatial equality of medical accessibility by adding new medical facility sites and adjusting the capacity of medical facilities by the number of beds. Finally, the optimized spatial equality values of medical accessibility under three travel time thresholds (15, 19, and 22 min) were compared to explore the effect of travel time thresholds of the accessibility time on the optimization effects.

## 2. Methods

Figure 1 shows the research framework of this study, which includes mainly three parts: (1) data preparation; (2) measurement of accessibility; (3) spatial equality optimization of accessibility.

In the process of preparing the data, we collected information on existing health facilities, such as their location and bed capacity. We used POI data and census data to create a grid-based representation of population distribution, and we calculated the travel time cost between each pair of supply–demand points by applying Gaode map's route planning API.

To measure medical accessibility, we classified medical facilities into two grades: high-level hospitals (Grade III comprehensive hospitals) and ordinary hospitals (Grade I and Grade II hospitals) and used the G2SFCA model to account for a facility's grade to assess accessibility.

To optimize the spatial equality of accessibility, we followed two steps: first, we established new medical facilities within the study area; second, we formulated a mathematical model for the optimization objective and then adjusted the number of beds for all facilities.

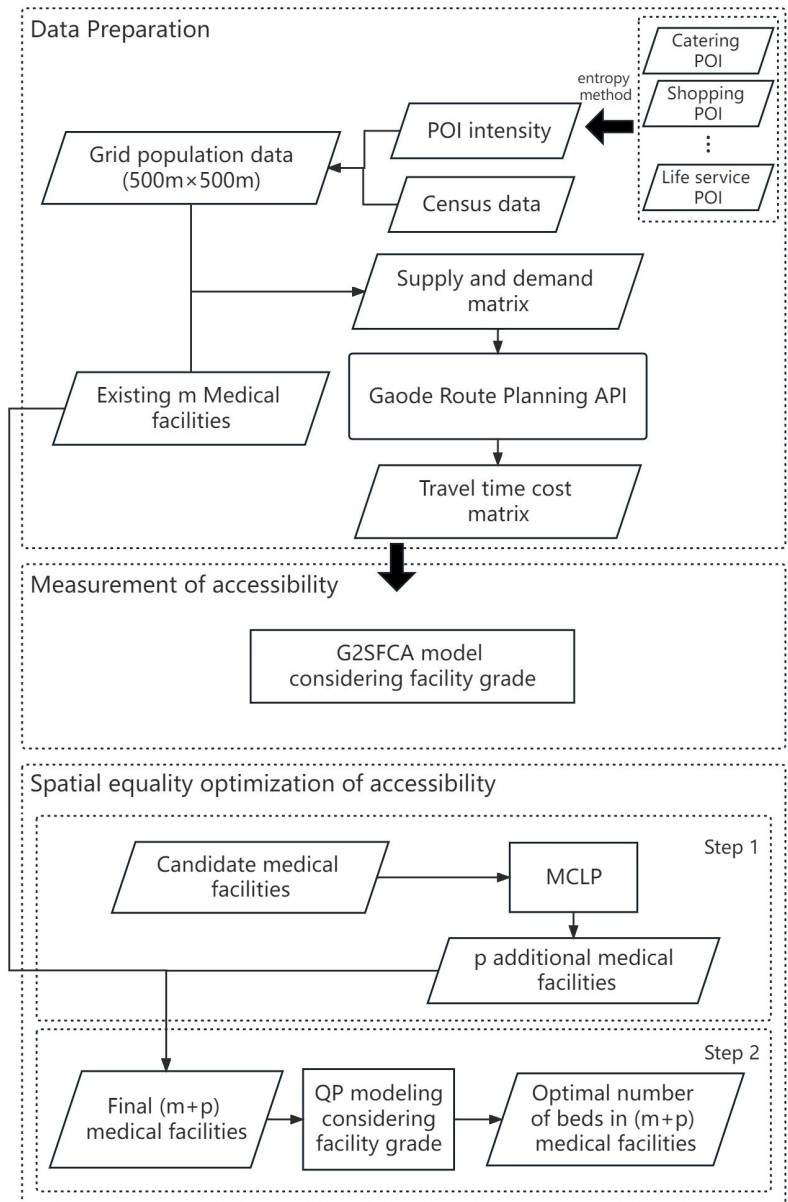

**Figure 1.** Research framework diagram.

*2.1. Measurement of Medical Accessibility*

Radke and Mu [56] first proposed a two-step floating catchment area (2SFCA). The idea of the algorithm is as follows. First, take supply point *j* as the center, calculate the total demand of all demand points within a certain threshold range, and then calculate the supply-demand ratio $R_j$ of supply point *j*. Step 2, take demand point *i* as the center, and count the sum of supply and demand ratios of all supply points within the threshold range. The calculation process is as follows:

$$R_j = \frac{S_j}{\sum_{k \in |t_{kj} \le t_0|} D_k} \tag{1}$$

$$A_i^F = \sum_{j \in |t_{ij} \le t_0|} R_j = \sum_{j \in |t_{ij} \le t_0|} \frac{S_j}{\sum_{k \in |t_{kj} \le t_0|} D_k} \tag{2}$$

The G2SFCA model conceptualizes the distance decay into various functions on the basis of 2SFCA. At present, the common distance decay functions mainly include

power, exponential, kernel density, and Gaussian functions. Because the distribution of the Gaussian function is similar to the normal distribution curve, which can better simulate the distance decay [57,58], this study used the Gaussian function as the decay function. In addition, considering that the service capacity of the Grade III hospitals is relatively higher than that of others, after referring to relevant studies and combining with the situation of the study area, the capacity weight of Grade III hospitals was set to 2, and the capacity weight of Grade I and Grade II hospitals to 1 [59]. Finally, the 2SFCA model can be rewritten as follows:

$$R_j = \frac{S_j \times W_j}{\sum_{k \in |t_{kj} \le t_0|} D_k \times G\left(d_{kj}\right)} \tag{3}$$

$$A_i^F = \sum_{i \in |t_{ij} \le t_0|} G\left(d_{ij}\right) \times R_j = \sum_{i \in |t_{ij} \le t_0|} \frac{G\left(d_{ij}\right) \times S_j \times W_j}{\sum_{k \in |t_{kj} \le t_0|} D_k \times G\left(d_{kj}\right)} \tag{4}$$

where $A_i^F$ is the spatial accessibility of demand point $i$; $R_j$ is the supply-demand ratio of supply point $j$; $S_j$ is the number of beds at supply point $j$; $t_{ij}$ is the travel time cost between the demand point $k$ and the supply point $j$; $t_0$ is the given travel time threshold, and different travel time thresholds corresponding to different medical coverage rates (both population coverage and area coverage). To investigate the difference in medical accessibility for different medical coverage scenarios, we set $t_0$ to 15 min, 19 min, 22 min (Table 1); $D_k$ is the demand of demand point $k$, measured by the population of demand point $k$; $W_j$ is the service capacity weight of supply point $j$; $G(d_{kj})$ is the Gaussian distance decay function. The calculation formula is as follows:

$$G(d_{kj}) = \begin{cases} \frac{e^{-\frac{1}{2} \times \left(\frac{t_{kj}}{t_0}\right)^2} - e^{-\frac{1}{2}}}{1 - e^{-\frac{1}{2}}} & (t_{kj} \le t_0) \\ 0 & (t_{kj} > t_0) \end{cases} \tag{5}$$

**Table 1.** Regional coverage and population coverage of medical services under different travel time thresholds.

| Travel Time Threshold | 15 min | 19 min | 22 min |
|---|---|---|---|
| Demand population coverage (%) | 91 | 95 | 98 |
| Demand area coverage (%) | 64 | 78 | 85 |

### 2.2. Spatial Equality Optimization of Accessibility

Optimizing the spatial equality of medical accessibility in this study involves two parts: (1) determining the optimal locations for additional facility points; (2) determining the optimal number of beds for each facility.

#### 2.2.1. Site Selection for New Medical Facilities

New medical facility sites should meet the requirement of covering the maximum range within their threshold (e.g., 10 min). This study used the maximum coverage location problem (MCLP) to select p medical facility sites from all candidate medical facility sites with *p* = 10. The method was implemented by "Location Assignment" in ArcGIS software. The demand weight was set to the area, and the problem type was set to "maximum coverage". For the setting of impedance interruption, it was found that the search threshold of 22 min can guarantee the coverage of 98% of the population in medical facilities, and a small increase in the search threshold did not increase the coverage rate significantly. Therefore, we set the impedance interruption to 22 min.

### 2.2.2. Optimizing the Number of Beds in the Facility

The optimization objective of this paper was to minimize the standard deviation ($\hat{A}$) of the accessibility index $A_i^F$ for all demand points in the study area. $\hat{A}$ is defined as below:

$$\hat{A} = \sqrt{\frac{\sum_{i=1}^{n}\left(A_i^F - \bar{A}\right)^2 D_i}{\sum_{i=1}^{n} D_i}} \tag{6}$$

which captures the total deviation of accessibility weighted by the amount of demand. Smaller $\hat{A}$ means higher equality. Total demand $\sqrt{\sum_{i=1}^{n} D_i}$ is a constant. Combining the definition of accessibility in Equations (3) and (4), the objective function of minimizing $\hat{A}$ can be transformed into the following:

$$min \sum_{i=1}^{n}\left(A_i^F - \bar{A}\right)^2 D_i = min \sum_{i=1}^{n}\left(\sum_{j=1}^{m} \frac{G\left(d_{ij}\right) \times W_j \times S_j}{\sum_{i=1}^{n} D_i \times G\left(d_{ij}\right)} - \bar{A}\right)^2 D_i \tag{7}$$

Note that the optimized number of beds $S_j$ ($j = 1, 2, \ldots, m$) of each medical facility is the variable to be solved and subject to $\sum_{j=1}^{m} S_j = S_t$. $S_t$ is the total number of beds in the study area, which includes the number of existing beds ($S_e = 99{,}406$) and the number of new beds ($S_n$). Due to the lack of policy documents related to the number of new beds, we assume that $S_n = 5000$. Therefore, $S_t = S_e + S_n = 104{,}406$.

Previous studies did not impose any restrictions on $S_j$, resulting in some unrealistic results, such as $S_j = 0$. This paper makes the following restrictions on $S_j$:

- Control the number of beds of existing medical facilities between $0.5S_j$ and $2S_j$
- Control the number of beds of new medical facilities above 20, because the "Basic Standards for Medical Institutions" stipulates that the number of beds in one hospital should be more than 20.

In Equation (7), $\bar{A}$ is the weighted average of $A_i^F$, equal to the ratio of total supply to total demand in the study area, such as:

$$\bar{A} = \frac{\sum_{j=1}^{m} W_j \times S_j}{\sum_{i=1}^{n} D_i} \tag{8}$$

Equation (7) can be transformed into solving convex optimization problems. Construct the following matrix:

$$G = \begin{bmatrix} G_{11} & G_{12} & \ldots & G_{1(m+p)} \\ G_{21} & G_{22} & \ldots & G_{2(m+p)} \\ \vdots & \vdots & \vdots & \vdots \\ G_{n1} & G_{n2} & \ldots & G_{n(m+p)} \end{bmatrix}, P = \begin{bmatrix} P_1 & 0 & \ldots & 0 \\ 0 & P_2 & \ldots & 0 \\ \vdots & \vdots & \vdots & \vdots \\ 0 & 0 & \ldots & P_{(m+p)} \end{bmatrix}, W = \begin{bmatrix} W_1 & 0 & \ldots & 0 \\ 0 & W_2 & \ldots & 0 \\ \vdots & \vdots & \vdots & \vdots \\ 0 & 0 & \ldots & W_{(m+p)} \end{bmatrix},$$

$$D = \begin{bmatrix} D_1 & 0 & \ldots & 0 \\ 0 & D_2 & \ldots & 0 \\ \vdots & \vdots & \vdots & \vdots \\ 0 & 0 & \ldots & D_n \end{bmatrix}, S = \begin{bmatrix} S_1 & S_2 & \ldots & S_{(m+p)} \end{bmatrix}^T, A = \begin{bmatrix} \bar{A} & \bar{A} & \ldots & \bar{A} \end{bmatrix}^T$$

where $G_{ij}$ refers to the gravitation between demand location $i$ and medical facility $j$, $P_i = 1/[\sum_{k=1}^{n} D_k \times G(d_{ik})]$ refers to the reciprocal of the total demand of service at medical facility $j$, $W_j$ is the capability weight of medical facility $j$, $W_j = 1, 2$, $D_i$ is the demand of location $i$, $S_j$ is the number of beds of medical facility $j$.

Defining matrix K = GP, Equation (7) can be rewritten as:

$$min\left(S^T W^T K^T DKWS - 2A^T DKWS + A^T DA\right) = min\left[2\left(\frac{S^T W^T K^T DKWS}{2} - A^T DKWS\right) + A^T DA\right] \tag{9}$$

The constant term $A^T D A = \bar{A}^2 \times \sum_{i=1}^{n} D_i$ and 2 can be dropped. Equation (9) can be further transformed into the quadratic programming (QP) problem in the following form:

$$min\left(\frac{x^T H x}{2} + q^T x\right) \tag{10}$$

Subject to

$$Cx \le b \tag{11}$$

$$Ex = d \tag{12}$$

In this study,

$$H = W^T K^T D K W,$$

$$q = \left(-A^T D K W\right)^T = -W^T K^T D A,$$

$$C = \begin{bmatrix} T_{2m \times m} & 0 \\ 0 & N_{p \times p} \end{bmatrix}, \ T_{2m \times m} = \begin{bmatrix} -1 & 0 & \cdots & 0 \\ 1 & 0 & \cdots & 0 \\ 0 & -1 & \cdots & 0 \\ 0 & 1 & \cdots & 0 \\ \vdots & \vdots & \vdots & \vdots \\ 0 & 0 & \cdots & -1 \\ 0 & 0 & \cdots & 1 \end{bmatrix}, \ N_{p \times p} = -1 \times I,$$

$$b = \begin{bmatrix} -0.5 \times S_1 & 2.0 \times S_1 & \cdots & -0.5 \times S_m & 2.0 \times S_m & -20 & \cdots & -20 \end{bmatrix}^T,$$

$$E = \begin{bmatrix} 1 & 1 & \cdots & 1 \end{bmatrix},$$

$d = S_t$, $x = S$, is the variable to be solved. This study used the convex optimization package (CVXOPT) in Python to solve x.

## 3. Study Area and Data

### 3.1. Study Area

The research area of this study is Guangzhou (Figure 2a), Guangdong Province, China. Guangzhou City located in 113°17′ E, 23°8′ N, is in the south of mainland China, the south-central Guangdong Province. With an area of 7434.4 square kilometers and a population of nearly 19 million, Guangzhou had a per capita GDP of 150,000 yuan in 2021. As the capital city of Guangdong Province, Guangzhou has always been the political, military, economic, cultural, scientific, and educational center of South China. Its medical level also ranks in the top places in China. At the end of 2020, the Hospital Management Institute of Fudan University released the "2019 China Hospital Ranking", among which, a total of nine hospitals in Guangzhou were listed in the top 100.

### 3.2. Data

#### 3.2.1. Medical Facility Data

The medical facility data used included both existing medical facilities and candidate medical facilities. Existing medical facility locations are available through the Web Services API (https://lbs.amap.com/, accessed on 1 October 2021). In previous studies, the number of beds in medical facilities was a commonly used indicator to describe the service capacity of medical facilities. In this paper, we also used the number of beds to measure the service capacity of medical facilities. The number of beds was obtained from the official websites of major hospitals and the 99 Hospital Database website (https://yyk.99.com.cn/, accessed on 1 October 2021). Since the "Basic Standards for Medical Institutions" stipulates that the

number of inpatient beds in "hospitals" should be more than 20, this study only focused on medical facilities with beds greater than or equal to 20. A total of 224 medical facilities were obtained after data cleaning, including 66 Grade III hospitals, with 500 to 3000 beds, and 158 Grade I and Grade II hospitals, with 20 to 500 beds (Figure 2b).

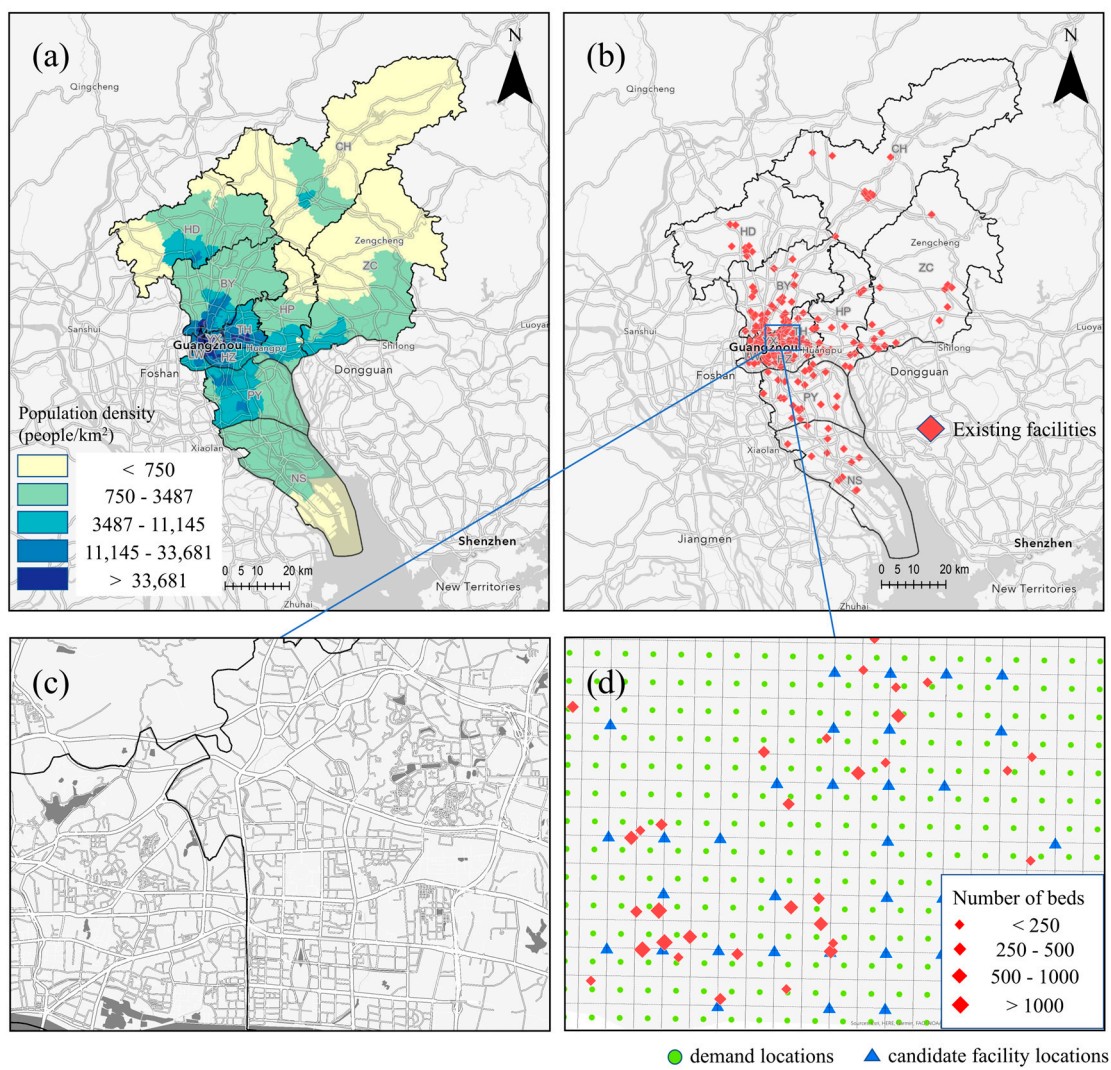

**Figure 2.** Study area. (**a**) Population density; (**b**) location of existing medical facilities (TH–Tianhe, YX–Yuexiu, HZ–Haizhu, LW–Liwan, BY–Baiyun, HP–Huangpu, PY–Panyu, HD–Huadu, CH–Conghua, ZC–Zengcheng, NS–Nansha); (**c**,**d**) show enlarged views of the same position.

Candidate medical facility locations are shown in Figure 2d. In this study, only candidate medical facility sites located within urban built-up areas were considered. Urban built-up areas were from the European Space Agency's land use data in 2020 (https:// viewer.esa-worldcover.org/worldcover/, accessed on 1 October 2021), which have a spatial resolution of 10 m. First, 1000 × 1000 m grids were established, and all grid intersections located in the built area were extracted. Then, the data were cleaned, removing sites in special locations such as airports. A total of 1378 candidate sites were obtained. These candidate sites were used to determine the location of new medical facilities.

### 3.2.2. Travel Time Cost Data

Most of the current studies estimate travel times based on the road network and a fixed speed approach, which leads to inaccurate estimation of travel time due to the neglect of actual road condition information and passing resistance. In a first-tier city like Guangzhou,

there are significant differences in road conditions and passing resistance in different areas. In central urban areas (such as Yuexiu district, Tianhe district, Liwan district), where traffic congestion is a common occurrence due to high population densities, using road network and fixed speed to measure travel time may lead to underestimation of results. The same situation also exists in some underdeveloped areas, although the impact is not as great as the central areas, but it cannot be ignored. In this study, we obtained the travel time cost in real-time by calling the Gaode route planning API (https://lbs.amap.com/, accessed on 1 October 2022). In order to obtain more detailed travel time costs, we first divided the research area into a number of grids. Generally, $1000 \times 1000$ m and $500 \times 500$ m grids are usually used for urban research and accessibility research [60–63]. However, using larger grids will increase the error of accessibility measurement [64]. so we divided the research area into a grid of $500 \times 500$ m, then, we extracted the center of each grid (green dots in Figure 2d) as the demand point and obtained its latitude and longitude, and the medical facility point (red dots in Figure 2d) as the supply point and obtained its latitude and longitude. Finally, we formed a supply and demand matrix with all supply points and demand points, and then called the Gaode route planning API to obtain the travel time matrix under the driving travel mode during off peak hours (3–5 pm and 9–12 pm).

### 3.2.3. Population Data

Studies have found that population density is highly correlated with points of interest (POI) density [65]. Therefore, this study obtained the population distribution map through the county level data of the seventh census and the POI data. The county-level data of the seventh population census is from the websites of the people's governments of all districts in Guangzhou, and the time is 2020. POI data is obtained through Web service API (https://lbs.amap.com/, accessed on 1 October 2021). For modeling purposes, the study area was divided into $500 \times 500$ m grids. All the POIs were classified into 10 categories according to their functions, namely catering, shopping, financial, education, life, entertainment, medical, hotel, residential, corporate, and enterprise. Considering the difference in the order of magnitude between different POI classes, we calculated the weight of different POI classes using the entropy method. The calculation process is as follows:

Step 1. The entropy of class $j$ POI is calculated as: $e_j = -\frac{1}{\ln n} \times \sum_{i=1}^{n} \left( \frac{Y_{ij}}{\sum_{i=1}^{n} Y_{ij}} \right) \times \ln \left( \frac{Y_{ij}}{\sum_{i=1}^{n} Y_{ij}} \right)$, where $Y_{ij}$ is the normalized value of the POI of class $j$ in the $i$th grid. Further, the weight of class $j$ POI is calculated as: $W_j = \frac{1-e_j}{\sum_{j=1}^{m} (1-e_j)}$. The number of POI in the $i$th grid can be calculated as $POI_i = \sum_{j=1}^{10} W_j \times Y_{ij}$.

Step 2. The proportion of the number of POI in the $i$th grid in the county (e.g., county k) is calculated. The formula is $POI_{i,k} = \frac{POI_i}{\sum_{j \in k} POI_j}$. Combined with the census data, the population in the $i$th grid is calculated by the formula $POP_i = C_k \times POI_{i,k}$; $C_k$ is the census number for the county k.

The calculated weight of POI is shown in Table 2. Considering the data availability, this study only focuses on the grids with a population greater than 0, which are called demand areas, while the non-population grids are considered as having no demand areas. Finally, 13,663 demand areas were obtained, as shown in Figure 3.

**Table 2.** Weight of each POI layer.

| Categories | Weight | Categories | Weight |
|------------|--------|------------|--------|
| Catering | 0.122 | Entertainment | 0.080 |
| Shopping | 0.146 | Medical | 0.087 |
| Financial | 0.084 | Hotel | 0.081 |
| Education | 0.090 | Residential | 0.081 |
| Life | 0.128 | Corporate and Enterprise | 0.101 |

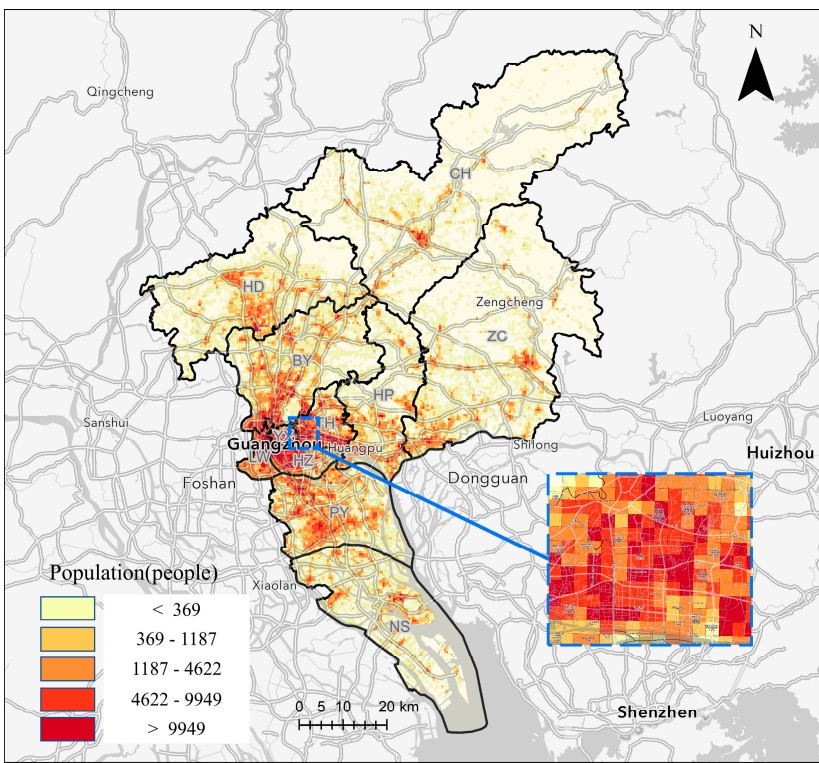

**Figure 3.** Population distribution.

## 4. Results

### 4.1. Measurement of Accessibility

In order to evaluate the stability and optimization effects of the optimization model, medical accessibility was measured and optimized under three different travel time thresholds (15, 19, and 22 min). The area and population coverage of medical services under different travel time thresholds are shown in Table 1, and the results are shown in Figure 4. Figure 4a−c shows the spatial distribution of access under different travel time thresholds. For ease of observation, the accessibility was classified into five levels based on natural breaks. For low travel time thresholds (Figure 4a), we found a large number of high-value clusters (>0.0126) in suburban areas, mainly Huangpu District and Panyu District. These clusters are not large and are mainly located in close proximity to medical facilities. The larger high-value clusters are located in the central city of Guangzhou, the main city of Zengcheng District, and the main city of Nansha District. Among these three larger high-value clusters, the central city of Guangzhou is characterized by a high level of demand and an equally high level of supply capacity. The main urban area of Zengcheng District and the main urban area of Nansha District have a relatively low supply capacity but similarly low levels of demand.

With greater travel time thresholds (Figure 4b,c), the regional differences in accessibility gradually decreased. It is noteworthy that a large number of high-value clusters located in suburban areas have significantly reduced in size or even disappeared, and the accessibility of these locations has been mainly transformed into the medium level. The reason is that with larger travel time thresholds, the range of medical facilities becomes larger and the demand for services increases, leading to a decrease in the ratio of supply to demand for individual medical facilities, and accessibility decreases when the services of new facilities are not available. However, we find that the two large high-value clusters in the central urban areas of Guangzhou and Nansha District still persist and have little morphological change, indicating that the change of travel time thresholds has no obvious impact on them, which is due to their strong medical service supply capacity and reasonable distribution pattern of medical facilities.

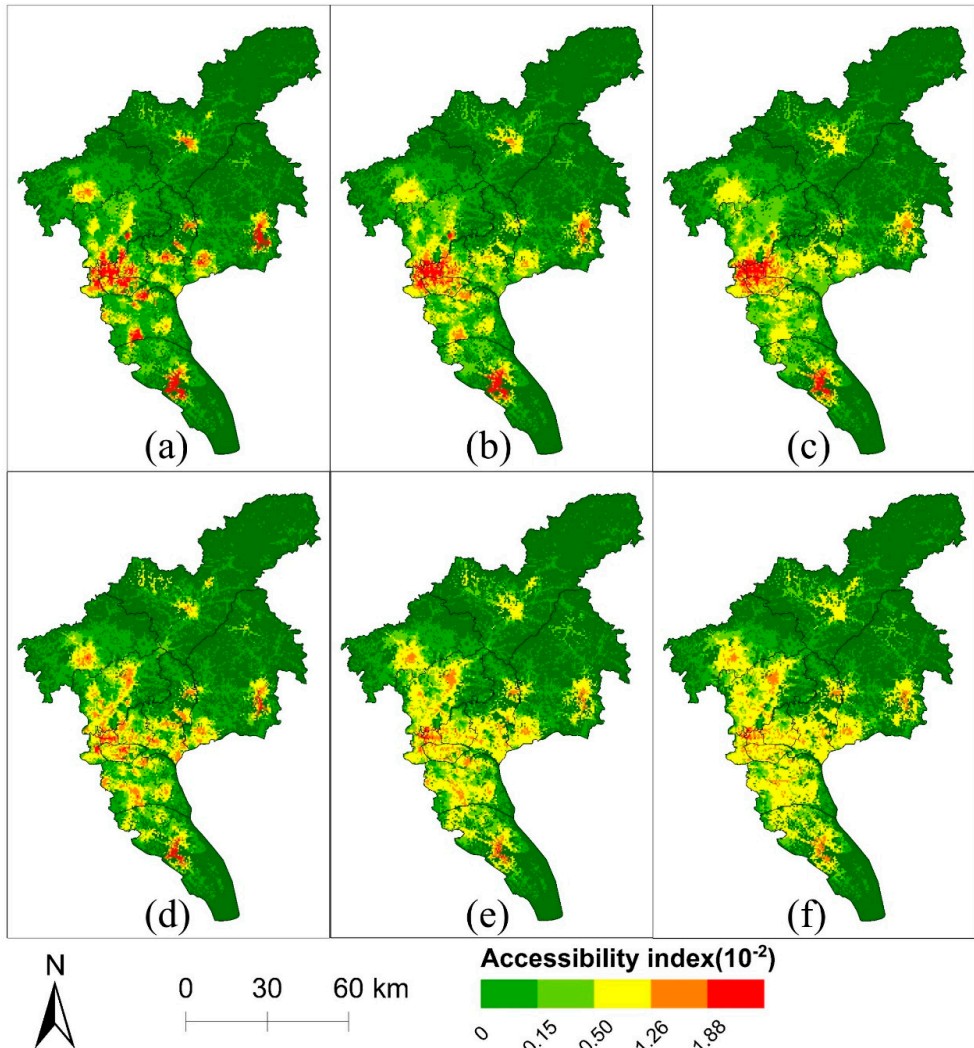

**Figure 4.** Spatial distribution of accessibility before and after optimization (((**a**,**d**): 15 min. (**b**,**e**): 19 min. (**c**,**f**): 22 min. (**a**–**c**) are before optimization and (**d**–**f**) are after optimization).

### 4.2. Equality Optimization Based on Accessibility

The accessibility distribution after siting locations and optimizing the number of beds (Figure 4d−f) shows that our model improves the spatial equality of accessibility. The optimization process significantly reduces the differences in medical accessibility, with clusters of very high accessibility largely disappearing and being replaced by clusters of medium and high accessibility, while areas of very low accessibility are somewhat reduced and transformed into low accessibility. However, at low travel time thresholds (Figure 4d), although optimization eliminates most of the very high accessibility clusters, it also generates high accessibility clusters at nearby locations, resulting in poor overall optimization. As the travel time thresholds increase (Figure 4e,f), the optimization effect gradually improves, leading to the disappearance of many high accessibility clusters and the expansion of the optimization scope. The possible reason for the reduction in the area of individual high accessibility clusters is that, as the travel time thresholds increase, more medical facilities can serve this area and more medical facilities enter into the optimization scheme, ultimately improving the optimization scheme. Overall, the optimization effect is better in the urban center, especially Guangzhou's central urban area, due to its dense distribution of medical facilities, compared to suburban areas.

### 4.3. Statistical Analysis of the Optimization Effects

To more accurately evaluate the impact of the optimization model on spatial equality, in addition to the standard deviation (SD) which we used to establish the objective function, we also calculated the maximum deviation (MD), mean absolute deviation (MAD), and coefficient of variation (CV). As can be clearly seen from Table 3, through the optimization of the model, under three travel time thresholds, all indicators have significantly decreased, indicating that the model has significantly improved spatial equality.

**Table 3.** Spatial equality of accessibility before (BO) and after optimization (AO).

| Indicator | Formula | 15 min | | 19 min | | 22 min | |
|---|---|---|---|---|---|---|---|
| | | BO | AO | BO | AO | BO | AO |
| MD | $\max\left(A_i - \bar{A}\right)$ | 0.1787 | 0.0847 | 0.0550 | 0.0237 | 0.0380 | 0.0149 |
| MAD | $\dfrac{\sum_{i=1}^n \left|A_i - \bar{A}\right| \cdot D_i}{\sum_{i=1}^n D_i}$ | 0.0078 | 0.0049 | 0.0067 | 0.0038 | 0.0062 | 0.0033 |
| SD | $\sqrt{\dfrac{\sum_{i=1}^n \left(A_i - \bar{A}\right)^2 \cdot D_i}{\sum_{i=1}^n D_i}}$ | 0.0111 | 0.0063 | 0.0089 | 0.0048 | 0.0080 | 0.0042 |
| CV | $\sqrt{\dfrac{\sum_{i=1}^n \left(A_i - \bar{A}\right)^2 \cdot D_i}{\bar{A} \cdot \sum_{i=1}^n D_i}}$ | 0.1145 | 0.0647 | 0.0910 | 0.0497 | 0.0824 | 0.0433 |

To further reveal the optimization performance, we studied the changes in the proportion of demand locations at different accessibility levels before (BO) and after optimization (AO) under different travel time thresholds. Figure 5 shows the optimization results at three travel time thresholds: t0 of 15 min, 19 min, and 22 min. In each figure, the first column shows the proportion of demand locations at each accessibility level before optimization, and the second column shows the proportion of demand locations at each accessibility level after optimization. These three figures reflect that: after optimization, the number of locations with very high (>0.0188) and very low (<0.0015) accessibility levels decrease significantly. The number of locations with medium accessibility level (0.0050–0.0126) rises sharply, and the increase is more pronounced when the travel time thresholds t0 is higher; low (0.0015–0.0050) accessibility levels show a decreasing trend while high (0.0126–0.0188) accessibility levels show a steady increase. At t0 of 22 min, medium accessibility level locations account for nearly 43%, which is a significant increase from 24% before optimization. Through optimization, most locations' accessibility levels move towards medium accessibility, but their movement is limited by travel time thresholds; larger thresholds result in larger overall movement while smaller thresholds result in smaller movement. The reason for this is that higher search thresholds ensure that more medical facilities participate in the optimization program to make more logical optimization results. These facts show that it is much more difficult for planning departments to achieve spatial equality in medical services when the acceptable travel time for most people is low. The higher the acceptable travel time for most people, the less difficult it is for planning departments to achieve spatial equality in medical services. However, in addition to considering acceptable travel times for most people, it is also important to consider people who need emergency medical services, and for such people, it is necessary to set a low travel time threshold. Therefore, from the planning department's perspective, it should not be problematic to set a higher travel time threshold, but this needs to take into account the interests and needs of various people.

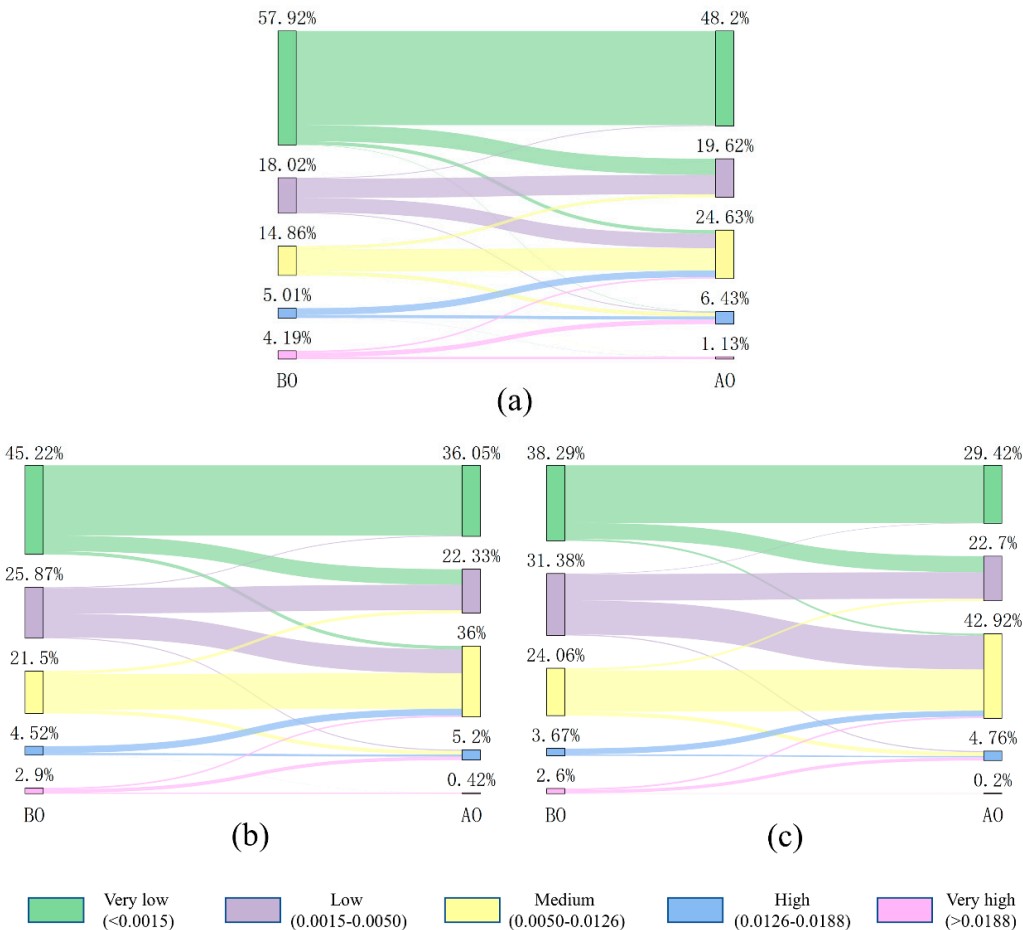

**Figure 5.** Change in the percentage of locations with different accessibility levels (BO: before optimization, AO: after optimization in different travel time thresholds of (**a**) t0 = 15 min, (**b**) t0 = 19 min, (**c**) t0 = 22 min).

The proportion of locations at different accessibility levels before and after optimization was further statistically analyzed at the municipal district scale (Figure 6). Before optimization (Figure 6a−c), the four districts with the best medical services in Guangzhou were Tianhe, Yuexiu, Haizhu District, and Liwan districts. More than 60% of locations in Yuexiu District have very high accessibility. The worst were Conghua, Huadu, and Zengcheng districts, where a very large proportion of locations had extremely low accessibility, mainly because these three districts have large areas but a sparse distribution of medical facilities. After optimization (Figure 6d−f), all districts' equality in medical accessibility improved to some extent, with a significant decrease in extremely low and high accessibility locations and a significant increase in medium accessibility locations. The difference in equality between different districts was also further reduced. However, there were varying improvements in different districts, and they were also affected by search threshold differences.

For each district, the changes in the proportion of locations for each accessibility level before and after optimization were calculated to describe the optimization effect of the model based on three travel time thresholds. Figure 7 shows the results of this analysis, with five points representing the proportion change (AO–BO) of locations at five accessibility levels for each district code. The horizontal axis represents the specific value of proportion change. The optimization effect is more evident when the five points are more scattered or when their connecting line varies more. Conversely, the optimization effect is less evident when there is less variation. From Figure 7a, when t0 = 15 min, Tianhe, Yuexiu, Liwan, and Haizhu districts, central districts in Guangzhou, have better optimization effects. However,

as t0 increases (Figure 7b,c), except for Yuexiu District, the improvement in the other three districts is not significant, with Tianhe District even slightly decreasing. However, Baiyun, Panyu, and Huangpu districts have significantly improved. Huadu, Nansha, Conghua and Zengcheng districts show no significant change. These facts indicate that the travel time threshold affects the optimization effect more significantly in Yuexiu, Baiyun, Panyu, and Huangpu districts than in Huadu, Nansha, Conghua, and Zengcheng districts. From the planning department's perspective, the first four districts are perhaps less difficult to optimize the equality of medical services than the last four districts. This suggests that the spatial allocation of medical facilities can be further improved in these latter four districts.

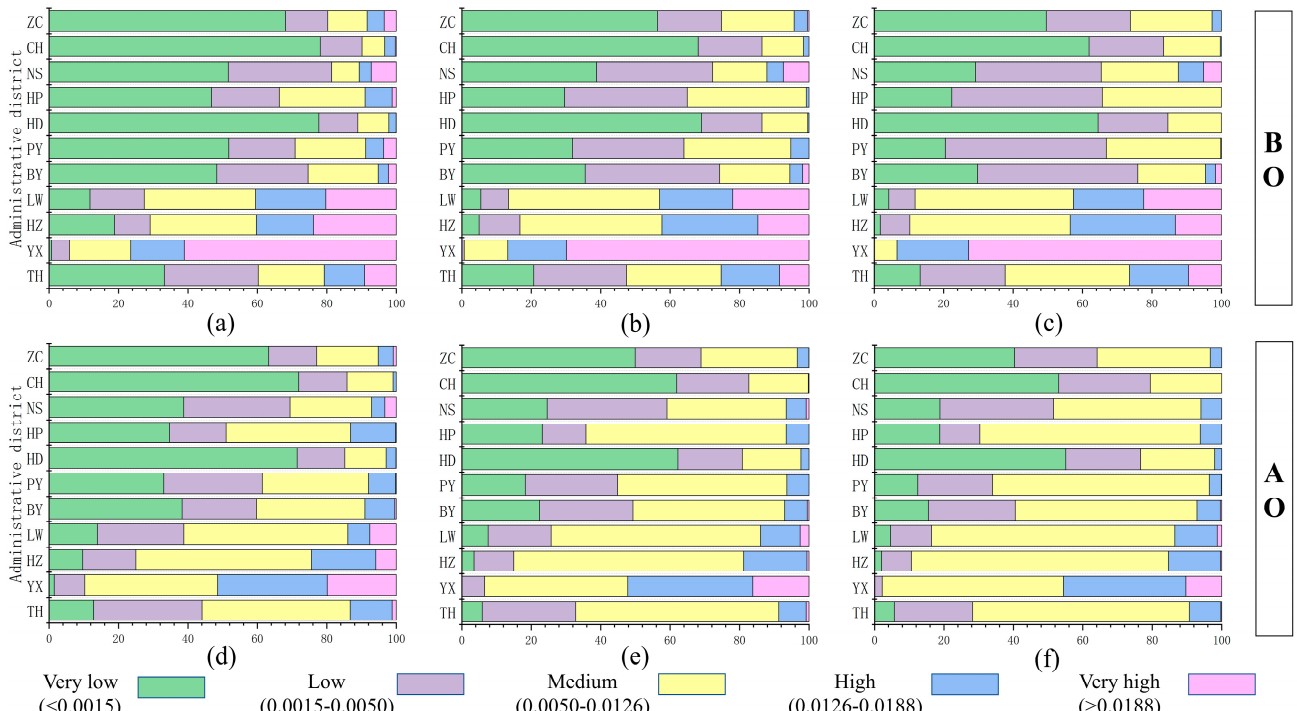

**Figure 6.** Regional statistics of location proportion of different accessibility levels (**a**,**d**): t0 = 15 min, (**b**,**e**): t0 = 19 min, (**c**,**f**): t0 = 22 min) before (BO) and after optimization (AO).

We analyzed the changes in the total number of beds in each district after model optimization when the travel time threshold was 22 min (Figure 7d). We observed that in Guangzhou's central urban districts (Tianhe, Yuexiu, Haizhu, and Liwan), except for a slight increase in the total number of beds in Tianhe, there was a varying degree of decrease in the remaining districts, resulting in a decline in overall medical accessibility. This was reflected in a significant decrease in the proportion of locations with high and very high accessibility levels, and a significant increase in the proportion of locations with medium accessibility levels. The peripheral districts (Baiyun, Panyu, and Huangpu), due to low overall accessibility before optimization, gained a large number of beds after optimization, resulting in an increase in overall accessibility. This was reflected in a significant decrease in the proportion of locations with very low and low accessibility levels, and a significant increase in the proportion of locations with medium accessibility levels. In the marginal districts (Huadu, Conghua, and Zengcheng), there were few beds obtained through optimization, with only a small improvement in overall accessibility.

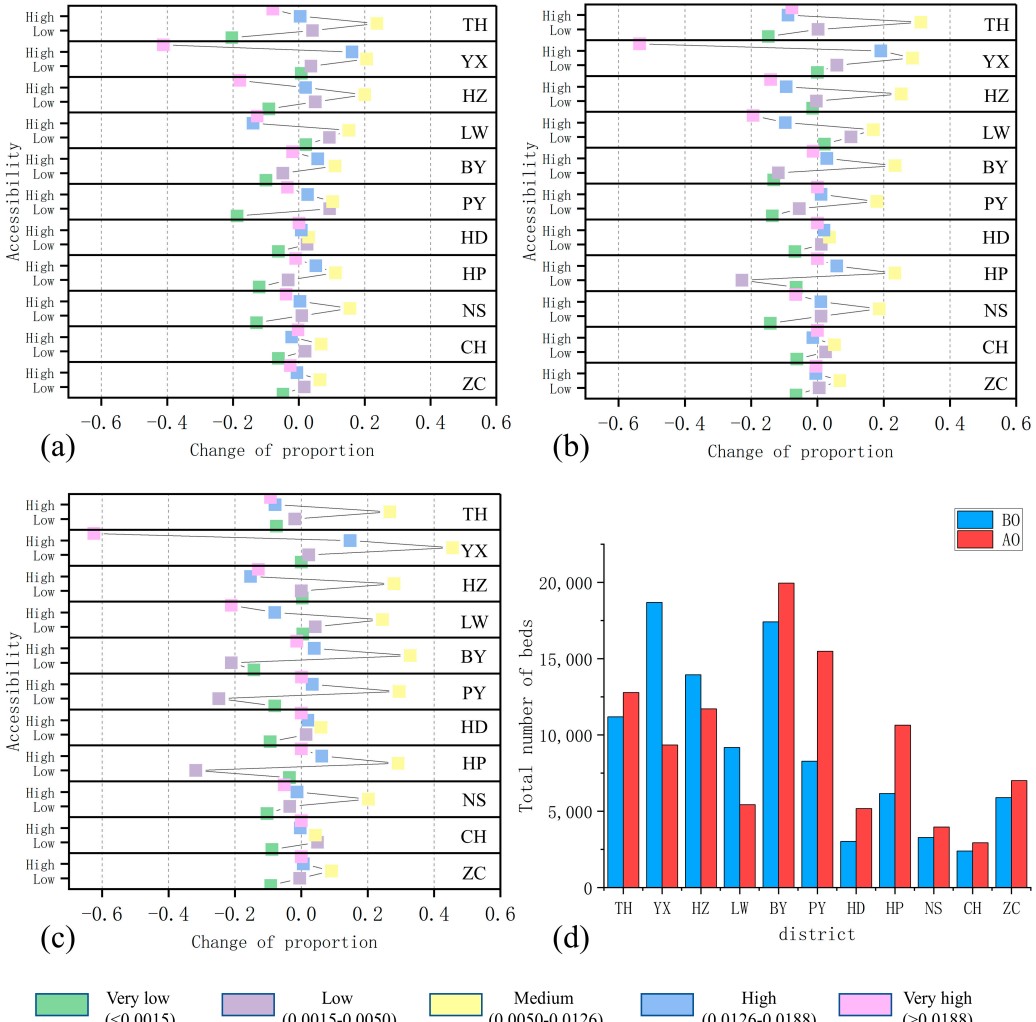

**Figure 7.** Change in the proportion of locations with different levels of accessibility at the municipal district scale ((**a**): t0 = 15 min, (**b**): t0 = 19 min, (**c**): t0 = 22 min), and (**d**): comparison of bed number before (BO) and after optimization (AO) when t0 = 22 min).

## 5. Discussion

### 5.1. Improvement of the Optimization Model

Previous studies on medical equality optimization have attempted to model the problem as a QP problem. However, these studies were based on the assumption that all facilities have equal service capacities. In fact, under China's hierarchical healthcare system, different levels of hospitals have different service capacity, and while many studies have considered facility-level differences in their accessibility measures, unfortunately, no study has taken into account facility-level differences in accessibility-based healthcare equality modeling. By introducing the service capacity weight matrix, this study establishes a QP optimization model that takes into account the facility level, bridging the gap of previous studies.

The range of facility capacity has not been restricted in previous studies. To address this problem, this study attempted to set the adjustment range of the number of beds to 0.5–2 times the original number of beds. A comparison of the optimization scheme generated before and after limiting the adjustment range of the number of beds (Figure 8) shows that if the adjustment range of the number of beds is not limited (Figure 8a), the resulting scheme is unreasonable, such as in the central urban area, where the number of beds of a large number of medical facilities is planned to be less than 20, including many Grade III hospitals. Referring specifically to Figure 8c, a large number of facility sites were

found near the x-axis, and the number of beds at these facility sites was eventually planned to be close to 0. A similar number of facility sites were found near the y-axis, and the number of beds at these facility sites dramatically increased. The above phenomenon is not possible in reality. After limiting the adjustment range of the number of beds (Figure 8b,e), the optimization scheme becomes more realistic. Comparing the number of beds at the same facility site before and after limiting the adjustment range of the number of beds (Figure 8d), it is found that there are few facility sites (red points) located near the line y = x. The number of beds planned of these facility sites is already optimal, and it is suggested that they can be configured according to that number of beds.

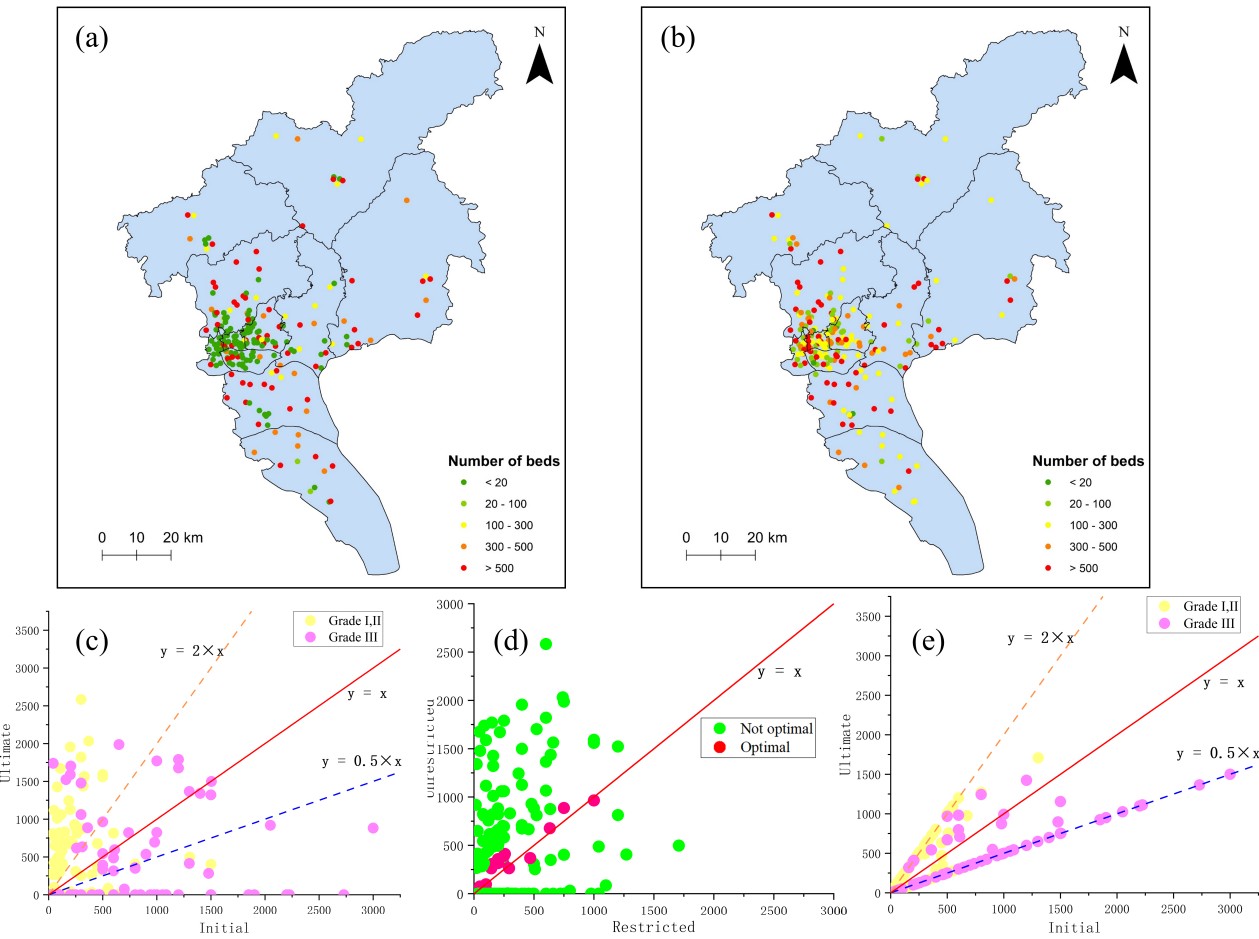

**Figure 8.** Impact of limiting the adjustment range of the number of beds on the optimization scheme (**a**,**c**): unrestricted, (**b**,**e**): restricted, (**d**): comparison of the planned number of beds in the same facility before and after the restriction.

In future models for optimizing equality in accessibility, adjustments can be made not only to bed numbers but also to medical facilities and medical personnel or adjusted simultaneously according to different weights. This requires that assessing medical service accessibility should be based on more features and comprehensively measure the service capacity of the medical facilities. Second, the optimization interval for the number of beds in each medical facility should be set more reasonably. Setting the limiting factor at 0.5 and 2 is still far from the reality. A more reasonable setting strategy should take into account the relevant policies, floor area, and bed utilization rate of each medical facility.

In addition, optimizing spatial equality in medical accessibility should not rely solely on adjustments by hospitals themselves but also coordinate with other departments such as transportation and urban planning [66]. The size of the optimization scope greatly affects optimization effects—good transportation conditions can provide a larger scope within a

certain search threshold which helps improve optimization effects. The location of new medical facility points also affects overall model optimization.

### 5.2. Differences in Optimization Effects among Municipal Districts

In this study, we measured the optimization effect by the changes in the proportion of demand location at each accessibility level. We found that the optimization effect varied across municipal districts and was closely related to the medical coverage rate of the municipal district. Urban fringe districts have low medical coverage rates and few demand locations that can be optimized, resulting in insignificant optimization effects. For example, in this study, Huadu, Conghua, and Zengcheng districts demonstrated this effect (Figure 9) and should build medical facilities and improve the transportation system in their low accessibility areas. For central urban areas, such as Tianhe, Yuexiu, Haizhu, and Liwan districts, the number of demand locations is relatively small and the medical coverage is saturated, so the optimization effect will not significantly improve the situation, because the model needs to devote resources to optimizing the accessibility of newly added demand locations in other municipal districts. In general, higher medical coverage means greater potential for optimization.

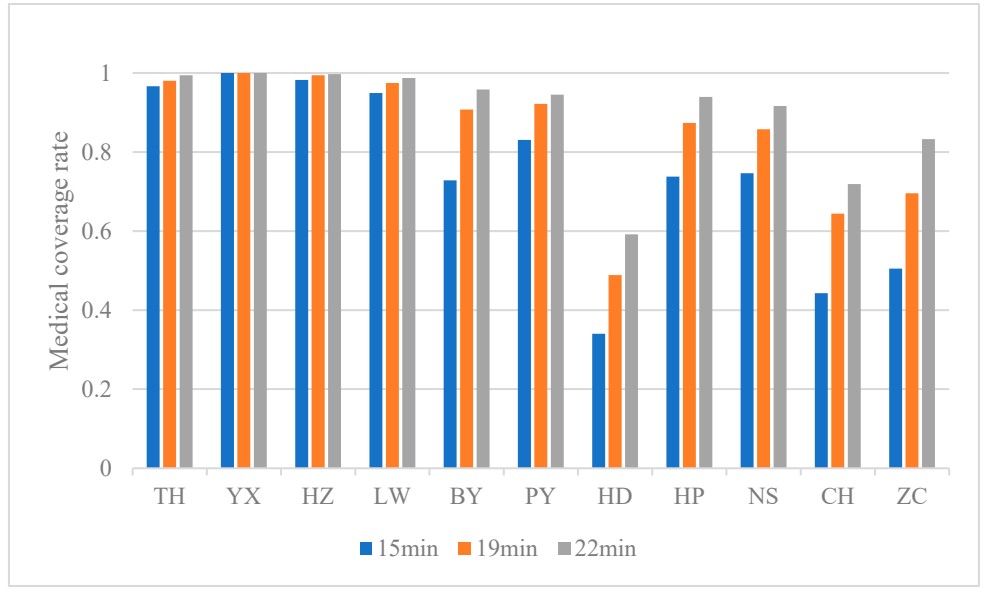

**Figure 9.** Medical coverage rate at the municipal district scale under different travel time thresholds.

However, this study also has some limitations. First, we only considered medical facilities with more than 20 beds as it was difficult to obtain accurate data for some small clinics, which were therefore excluded. This may have led to an underestimation of medical accessibility in some areas (Figure 4), especially in underdeveloped areas with lower population density. These areas are the ones that benefit from the equality optimization of medical services because local medical resources are neglected, so the optimization process allocates additional medical support to these areas. Second, the maximum search threshold in this study was 22 min, which only guarantees 98% of the demand coverage. Therefore, the optimization mechanism of the model can only be explained at 98% of the population coverage. Third, the model was designed without taking into account the important factor of socioeconomic composition. The spatial inequality of medical services is essentially the spatial inequality of socio-economic development, so until the issue of spatial inequality in socio-economic development is resolved, it is difficult to completely resolve the issue of medical spatial inequality. These aspects should be areas of future research improvement.

## 6. Conclusions

This study examined the spatial equality of medical accessibility in Guangzhou City, China. It employed the maximum coverage location problem (MCLP) to select optimal locations for new medical facilities that maximize the coverage area. Then, it incorporated a service capacity weight matrix and formulated a QP model that aims to minimize the standard deviation of accessibility across all demand locations. To investigate the differences in the optimization effects of the model under different travel time thresholds, we performed three scenarios based on varying travel time thresholds.

The case study led to several conclusions. First, to optimize the number of beds in facilities, it is important to consider the differences in hospital levels, which can reduce the underestimation of service capacity of high-level hospitals and present a more accurate picture of accessibility distribution in the study area. The model is effective in improving spatial equality of medical accessibility, as it increases accessibility for locations with extremely low or low levels (mainly peripheral districts) and decreases accessibility for locations with extremely high or high levels (mainly central urban areas). However, relocating medical resources may increase the difficulty of accessing medical care for residents in central urban areas; whereas, from the perspective of Guangzhou's planning department, it could improve medical services in a wider area, narrow regional disparities and align with the goals of the 14th Five-Year Plan.

Second, the model achieves better optimization at higher travel time thresholds. This suggests that when the majority of residents can accept longer medical travel times, the easier it is for the planning department to achieve spatial equality in medical care, and the more reasonable the optimal solution obtained by the model.

Third, the impact of travel time threshold on optimization effect varies among different districts, mainly reflected in the magnitude of improvement. With increasing travel time threshold, the improvement magnitude for the model optimization effect are as follows: peripheral districts (Baiyun District, Huangpu District, Panyu District) > central urban areas (Tianhe District, Haizhu District, Liwan District) > edge districts (Huadu District, Conghua District, Nansha District). Among them, edge districts have the least obvious optimization effect and a very small improvement magnitude, indicating that there is still room for further improvement in the number and distribution of medical facilities. It is recommended to add more medical facilities and improve the transportation system at the same time.

**Author Contributions:** Conceptualization, Mingkai Yu, Yingchun Fu and Wenkai Liu; Data curation, Mingkai Yu; Formal analysis, Mingkai Yu; Funding acquisition, Yingchun Fu; Methodology, Mingkai Yu; Project administration, Yingchun Fu; Resources, Mingkai Yu; Software, Mingkai Yu; Supervision, Yingchun Fu and Wenkai Liu; Validation, Mingkai Yu, Yingchun Fu and Wenkai Liu; Visualization, Mingkai Yu; writing—original draft preparation, Mingkai Yu; writing—review and editing, Yingchun Fu and Wenkai Liu. All authors have read and agreed to the published version of the manuscript.

**Funding:** This research was funded by the National Natural Science Foundation of China (NSFC), grant number 42071399, and the Marine Economy Development Foundation of Guangdong Province, grant number GDNRC[2022]21.

**Data Availability Statement:** Data or models that support the findings of this study are available from the corresponding author upon reasonable request.

**Acknowledgments:** The authors thank the editors and anonymous reviewers for their insightful comments and constructive suggestions.

**Conflicts of Interest:** The authors declare no conflict of interest.

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
