# Peer review of "An Optimization Method for Equalizing the Spatial Accessibility of Medical Services in Guangzhou"

_ijgi, doi:10.3390/ijgi12070292_

Round 1
Reviewer 1 Report
This paper describes an optimization method for the equalization of accessibility to medical services. It contains five sections. The first section is an introduction explaining the need for equality in the accessibility to medical services, and describing existing research on health service planning, and more precisely on tools to measure and improve spatial equality in accessibility. The second section describes Guangzhou, the study area of the paper, then describes how data (on medical facilities, number of beds, population densities, travel times) were found or computed, and finally gives a detailed workflow of the study and mathematical details on the optimization methods used to determine optimal sites for new facilities and optimal number of beds for existing and new facilities. Section 3 gives the results of the study in Guangzhou, and section 4 explains how the model improves on previous studies, and which modifications could yield further improvement. Section 5 concludes and summarizes the main findings of the study.
This article improves on existing methods by taking into account facility grade differences and facility capacities in the computation of accessibility (giving more weight to grade III facilities), setting more realistic constraints on facility capacities after optimization, and exploring the effects on time travel thresholds on the effects of optimization. The method, the data used, and the results of the study are described with sufficient detail, and the paper is very clear and understandable in general. In the discussion part, the authors adequately discuss the advantages of their approach, and particularly the need to put constraint on facility capacities.
My main remark would bear on how the author study the influence of travel time on the optimization problem. Throughout the paper, the authors' findings are that higher time travel thresholds enable better optimization (l. 378 for example). I am wary of such statement ; it is true that optimization is objectively 'easier' for higher thresholds, leading to lower inequality indicators, but the tone of the analysis could lead the reader to understand thathigher thresholds are inherently better. From a planning perspective, the most relevant threshold depends less on the quality of the optimiation than on external factors such as travel time accepability for populations, or how quick people must be taken care of in case of emergency. The authors should perhaps be clearer in aknowledging that.
More specifically, the authors do not explain why they chose 15min, 19 min and 22 min as possible time thresholds. are they arbitrary, where they chosen considering external constraints, or with an analysis of the data and of the overall coverage they yield?
In section 4, the authors explain how the constraints they put on the capacities of facilities played a large role in the results of the study, as a lot of facilities either reached their maximum or their minimum constraints. These constraints are necessary to make the optimization more realistic, but I still feel that the factors 2 and 0.5 are very lenient, because doubling or especially dividing by a factor of two the capacity of a facility is not always doable realistically. When the authors describe further improvements on their method, setting more adequate constraints, or assigning costs to large changes of existing facility capacities could be mentioned.
Here are a few more minor remarks:
Subsection 2.3 is important enough, in both size and content, to be a full section on its own (possibly before the description of the study area and the way data were obtained)
Figure 1 : text of map b) is readable, but very small.
map c) has some text that is not readable ; it could be removed, or relevant information should be made more readable.
Caption of Figure 1 : The list of cities refers to map b), so it should appear after the description of b), not at the end.
Here are a few corrections that could be made:
l.25-26 : this sentence has no verb and is difficult to understand
l.55 : there is lack of → there is a lack of
l.204 : Research framework diagram of this study : something seems to be missing in this sentence
l.237-241 : in this paragraph, indices do not appear as indices but have the position and size of normal text, which is inconvenient (but still understandable). The same problem appears several times in the paper
Equation (6) (l.263) : the quantity written A_i^F in the remainder of the paper is only written as A_i here
l.273 : we assume that S_e=5000 : it should be S_n=5000
l.278-279 : stipulates that the number of "hospitals" should be more than 20. : the number of beds in one hospital should be more than 20.
English grammar and vocabulary is generally good. Below are suggestions bearing only on grammar/vocabulary and sentence construction:
l.25-26 : this sentence has no verb and is difficult to understand
l.55 : there is lack of → there is a lack of
l.204 : Research framework diagram of this study : something seems to be missing in this sentence
Reviewer 2 Report
Thank you for giving me this opportunity to read the manuscript entitled "An optimization method for equalizing the spatial accessibility 2 of medical services in Guangzhou". The paper presents a comprehensive methodology that includes data preparation, measurement of accessibility, and spatial equality optimization of accessibility. The authors have used the two-step floating catchment area (2SFCA) method to measure medical accessibility and have classified medical facilities into two grades: high-level hospitals and ordinary hospitals. Overall, the paper presents an interesting and the methodology is sound, but there are some limitations that need to be addressed before it is considered to be published.
1. The maximum search threshold in this study was 22 minutes, which only guarantees 98% of the demand coverage. This threshold poorly explains the optimization mechanism of the model and may not fully represent the accessibility of medical services for all residents.
2. The existing optimization model for equality of medical services does not take into account facility grade differences. This could potentially lead to an inaccurate representation of the accessibility of medical services.
3. There is no reasonable limit in the optimization range of facility capacity which could result in the possible situation of zero capacity, which is not realistic in a practical context.
4. More exploration should be added on how different travel time thresholds affect the optimization effects and their influencing factors. The lack of such information could limit the understanding of how travel time impacts accessibility and the effectiveness of the optimization model.
5. Lines 61-63: “Accessibility, a concept borrowed from the field of 61 transportation planning, has also been widely used in the field of urban public services 62 [23–25].”: a paper titled "Observed inequality in urban greenspace exposure in China" could be added as a reference to support the statements here.
Minor editing of English language required
Reviewer 3 Report
Equity in the spatial access to medical services is an important urban planning issue that burgeoning studies are addressing. Many models have emerged to date for evaluating equitable spatial access of medical services. The model introduced in this manuscript is one of them. This is indeed an important addition to the array of models since it adds new dimensions. However, I have a few comments.
1. In the introduction section, the author mentions two concepts “efficiency” and “equality”. It will be very easy to follow this article if it is clearly mentioned what the authors mean by “efficiency” and “equality”. For example, a simple definition of “efficiency” can be included after line 43, after the authors mention the term.
2. The author does a thorough review of articles in the introduction. However, all of the models mentioned in the review are based in China. It will be helpful if the authors can include some information regarding what models are being applied in other countries to deal with similar issues.
3. Also, the models are quite dated. New models are emerging addressing similar issues. For example: Liu, L., Zhao, Y., Lyu, H., Chen, S., Tu, Y., & Huang, S. (2023). Spatial Accessibility and Equity Evaluation of Medical Facilities Based on Improved 2SFCA: A Case Study in Xi’an, China. International Journal of Environmental Research and Public Health, 20(3), 2076. Discussion on some emerging models and where do the author’s developed model stand in comparison to the new ones can be included.
4. The authors include Figure 1(a) in the manuscript. However, they do not mention the figure anywhere in the text. I think the figure can be excluded from the manuscript since it has not been used.
5. More information on travel time cost data is desired. I understand that the data was readily available. However, some description of how the travel time cost is estimated is important to understand. For example, was all different commuting modes were used in the calculation or only one. This explanation makes a significant difference in understanding the accessibility context in this manuscript.
6. Also, some additional context on the status of transportation in different parts of the city can be included. I think such context is critical for understanding accessibility.
7. The authors mention that smaller medical services were not included in the research due to data availability issues. However, low accessibility in the low-population density areas found in the research is not surprising since those areas may have been adequately served by smaller medical services. The authors can think of including an explanation of the implication of ignoring smaller medical services in the overall findings.
8. No accessibility issue can be effectively addressed until socioeconomic differences are considered. The model does not consider socioeconomic components when devising the model. The authors can acknowledge the importance of including socioeconomic variables in future versions of the model.
